# A Peculiar Case of Ossicular Chain Fixation and Enlarged Vestibular Aqueduct

**DOI:** 10.3390/children10020360

**Published:** 2023-02-11

**Authors:** Davide Brotto, Marzia Ariano

**Affiliations:** Section of Otorhinolaryngology-Head and Neck Surgery, Department of Neurosciences, University of Padova, 35128 Padova, Italy

**Keywords:** conductive hearing loss, ossicular chain fixation, enlarged vestibular aqueduct

## Abstract

We present the case of a patient treated as affected by conductive hearing loss due to recurrent otitis, then as a juvenile otosclerosis, who was finally diagnosed as affected by bilateral ossicular chain fixation and enlarged vestibular aqueduct by means of cone-beam CT.

## 1. Introduction

Conductive hearing loss is an extremely frequent condition during childhood, especially in the first years of life, when anatomical, immunological, and social conditions favor the high incidence of recurrent otitis media [1].

This condition typically causes a conductive hearing loss that might have a tremendous impact on the child’s development since an impaired hearing function can cause delays in speech development and mid-to-long-term consequences on the child’s quality of life [2].

The extremely high prevalence of this condition in the first years of life makes the correct characterization of congenital hearing loss very challenging, especially in complex and fragile patients (i.e., patients with multiple disabilities, with poor cognitive performance or affected by autism spectrum disorders). Indeed, the occurrence of the combination of different rare conditions might mimic this extremely frequent pathology, and only patient performance in behavioral audiological tests is sometimes able to produce a correct diagnosis of hearing loss.

All of the conditions that might cause conductive or mixed hearing loss frequently require multiple examinations to reach a good characterization of the hearing deficit. The audiological evaluation is frequently followed by radiological imaging and, in some cases, by genetic testing. Only the collection of all these data can lead to an accurate final diagnosis.

Herein, we report the case of a patient diagnosed and treated first as affected by recurrent otitis media, then as affected by otosclerosis. In the end, it was recognized to be a case of congenital ossicular chain fixation and concomitant enlarged vestibular aqueduct (EVA).

## 2. Detailed Case Description

A 13-year-old boy presented to our tertiary referral center with mixed mild–moderate bilateral hearing loss, worse on the left side (Figure 1).

Clinical history indicated previous surgeries for lingual frenectomy and multiple tympanostomic tubes. He also underwent speech rehabilitation for speech delay. Apart from borderline cognitive function, no other major diseases were reported.

Upon clinical ENT examination, otoscopy was negative for inflammatory or infectious middle ear pathologies.

The information concerning previous audiological examinations was scattered. In different centers, the bilateral hearing loss was deemed to be conductive in the first years of life, but as the boy grew up, the better-characterized mixed hearing loss was interpreted as juvenile otosclerosis, and stapedotomy was performed on his right side. Air conductive hearing aids were suggested, although the patient, especially after surgery, preferred to use it only on his left ear.

At this point, the patient was referred to our center to obtain a second opinion, given the unsatisfactory hearing improvement after surgery. A cone beam computed tomography (CBCT) of the temporal bone was performed, revealing bilateral enlarged vestibular aqueducts (EVAs) associated with a malformation of the ossicular chain, which appeared bilaterally fixed to the upper lateral wall of the middle ear (Figure 2). The vestibular aqueduct diameter at the midpoint resulted to be 1.74 mm on the right and 2.45 mm on the left.

## 3. Discussion

Otitis media, with its recurrent course in most patients, is the most frequent cause of conductive hearing loss in the first years of life: it is estimated that in the post-pneumococcal conjugate vaccine era, about 60% of children aged less than 4 years have ≥1 episode of acute otitis media [1]. Clinicians are well aware that the persistence of the pathology not only affects the structures of the middle ear but may also cause hearing impairment and potential communicative issues [1].

In the acute phase of the disease, the most common symptoms are very non-specific and may include fever, otalgia, headache, irritability, cough, rhinitis, listlessness, anorexia, vomiting, diarrhea, and pulling at the ears [3]. These, along with an accurate examination of the tympanic membrane, allow for proper diagnoses, and, most importantly, differentiate a suppurative ear infection from a viral one, thus avoiding unnecessary antibiotic treatment. For a good clinical examination of a patient, it is first mandatory to have an otoscope with a fresh bulb and a good power source; the tympanic membrane must be fully visible and not hidden by cerumen. The most common presentation is a bulging, cloudy, immobile tympanic membrane, while an isolated erythema of the eardrum is often the result of viral infection, crying, or attempts to remove the cerumen [4,5]. In the first phases of the disease, in uncomplicated cases, the physician could suggest the use of nose drops to improve eustachian tube ventilation and pain therapy [6]. The most common organisms isolated from middle ear fluid are *Streptococcus pneumoniae, Haemophilus influenzae, and Moraxella catarrhalis* [7], and antibiotics can be administered to reduce the rate of complications such as mastoiditis, facial nerve paresis, and labyrinthitis: these drugs are recommended in all children younger than six months, in those between six months and two years if the diagnosis is certain, and in children with severe infection [3]. For recurrent cases, in the presence of an abnormal ear examination, it is often proposed to use a more active approach in terms of the patient, typically by placing a tympanostomy tube in the affected ear [8], even if some authors disagree on its efficacy in preventing recurrences when compared to antibiotic therapy [9]. Moreover, although otitis media is often treatable and may resolve spontaneously without complications, in some cases, it can be associated with hearing loss and life-long sequelae, especially in developing countries [10]. In fact, to prevent this condition, for the more severe acute cases, especially if a known congenital middle ear anomaly is present [11], mastoidectomy should be considered as soon as possible before conductive hearing impairment occurs [12].

In children with persistent hearing loss following tympanostomy tube placement, an accurate characterization of the type and the side of the hypoacusis should be made: if bilateral hearing loss is present, imaging and genetic testing should be considered (such as what happened regarding our patient), to identify a possible syndrome. Patients with mixed hearing loss should be evaluated for middle/inner ear anomalies [13].

The difficulty in addressing the possible hearing loss in children is that multiple non-disease-related factors can impair pediatric audiological evaluation (for example, the patient’s reduced collaboration). Some patients might also be affected by specific conditions that impair the diagnostic process: those factors could be represented by other comorbidities that need to be addressed before the hearing management or by the presence of cognitive and behavioral issues which might impair the audiological evaluation.

Based on the age of the patient, the diagnosis of hearing loss might be reached using different techniques: for infants under the age of 6 months, auditory brainstem response or otoacoustic emissions; for children up to two years of age, behavioral audiometry, and after this, conditioned play audiometry [14]. Although the abovementioned are very useful techniques for assessing the presence of some kind of hearing impairment, they are obviously less accurate compared to pure tone and speech audiometry in reaching a clear diagnosis of hearing loss in terms of the degree, type, and impact on the overall child development. Moreover, unfortunately, in some patients, it is not possible to perform all of the tests that would be considered useful for the clinician. Consequently, reaching a precise diagnosis is not so simple.

When the first-line treatments continue to fail in treating conductive hearing loss, an in-depth evaluation should be performed, adding both objective audiological tests (such as tympanometry, cochlear stapedial reflexes, and auditory brainstem responses) and radiological imaging (especially with computed tomography in the case of mixed hearing loss).

It is recommended to perform computed tomography (CT) using both the Axial and the coronal planes because they are complementary in visualizing all of the different ear structures [15]. The most common CT findings in an ear affected by chronic otitis media, which might predict hearing impairment, are the presence of soft tissue density in the antrum and in the round window niche [16]. In more severe cases, bone erosion, exposed dura, ossicular chain discontinuity, and facial canal dehiscence might also be detected [15]. The CT should be considered the first-line imaging tool in the case of persistent mixed hearing loss. This condition may be due to the presence of congenital or acquired pathology that can be visualized with X-ray based techniques, while magnetic resonance imaging is less useful. The pathological processes can affect the middle ear with congenital malformations of the ossicular chain (i.e., anomalies in the shape of the ossicles, fixation to the cavum tympani walls, or malformed junction between the ossicles).

If we are talking about rare causes of conductive congenital hearing loss, congenital middle ear anomalies, defined as malformations of the auditory ossicles of any type, should be considered. The ossicular chain has an embryological origin from the branchial arches (focusing on the incus, it derives either from the first branchial arch alone or from both the first and second branchial arches) [17], and malformations can occur at any stage of this delicate development. The conductive hearing loss usually found is mild to moderate in degree, and additional external ear abnormalities could be present, facilitating the patient’s referral to tertiary centers and a prompt audiological workup for the detection of anomalies of the pinna. Middle ear malformations are predominantly sporadic, but they may also be part of a syndromic condition in about 25% of cases [11]. Possible genetic anomalies determining these conditions have yet to be clearly defined. Some papers have suggested the probable autosomal dominant [18] or X-linked-dominant [19] transmission of the genes associated with ossicular chain malformations. Despite the extremely rare occurrence of this kind of malformation and the small number of articles present in the literature, Cremers and Teunissen proposed a classification dating back to 1993 [20]:

Class 1: ears with congenital isolated stapes ankylosis.

Class 2: ears with congenital stapes ankylosis and a congenital anomaly of the ossicular chain.

Class 3: ears with congenital anomalies of the ossicular chain and at least a mobile stapes footplate.

Class 4: ears with aplasia or severe dysplasia of the oval window or round window.

Usually, class 3 anomalies comprise the majority of middle ear anomalies, while class 4 is the one with the worse post-surgical hearing outcomes [21]. Our patient presented with stapes ankylosis as well as the fusion of the ossicular chain with the lateral wall of the tympanic cavity, thus making it possible to consider his malformation a class 2. Additionally, he showed poor cognitive performance and the presence of another rare inner ear anomaly, meaning a bilateral enlarged vestibular aqueduct (EVA): both his ear malformations were clearly evidenced by cone beam computed tomography (CBCT).

Other causes of juvenile hearing loss identifiable with CT are progressive diseases of the middle ear that can cause fixation, decalcification, or fractures of the ossicles (i.e., chronic otitis media or cholesteatoma). These conditions cause mixed hearing loss when combined with inner-ear-derived sensorineural hearing loss, but in some cases, the progression of the abovementioned pathologies within the otic capsule, such as a chronic infection/inflammation or cholesteatoma, can cause a mixed hearing loss with a single etiology. CT can also visualize the presence of otosclerosis, characterized by osteo-rarefaction and otospongiosis at the fissula ante fenestram with frequent inner ear involvement [22].

Additionally, some inner ear malformations may cause mixed hearing loss. It is crucial to identify these anomalies because the management and course of hearing loss may be extremely different among these conditions. For example, incomplete partition type 2, incomplete partition type 3, or cochlear hypoplasia may show similar audiological characteristics, but the anatomical features are extremely different.

In our patient, a concomitant EVA was revealed by CT examination.

EVA is the most commonly identified malformation of the temporal bone, and it is reported to be the most common inner ear malformation associated with congenital hearing loss [23]. EVA can be found in up to 32% of pediatric patients with non-syndromic sensorineural hearing loss (SNHL) [24].

The diagnosis of EVA is based on high-resolution, fine-section CT scans of the temporal bone. According to Valvassori and Clemi’s criteria, EVA can be diagnosed if the diameter of the aqueduct is ≥ 1.5 mm at the midpoint or ≥ 2 mm at the operculum, as in our case [25]. In some cases, T2-weighted magnetic resonance imaging could be needed to obtain information related to membranous structures, such as the endolymphatic duct and the endolymphatic sac [26].

Although an EVA could be observed alone, it may also be associated with cochlear and modiolar defects (in this case, the diameter is reported to be larger than in an isolated EVA), even if the literature is scarce regarding its association with middle ear defects, such as the case of our patient [27].

Moreover, clinically speaking, EVA can be an isolated finding, or it can be associated with specific syndromes, such as Pendred, Waardenburg, or branchio-oto-renal syndrome [23].

The course of hearing loss is often progressive, starting from normal/almost normal hearing thresholds and worsening over time towards a conductive/mixed type, ranging from mild to profound. Hearing loss can be conductive (caused by a limitation of the stapes movement due to the abnormally high inner ear fluid pressure and by the “third window effect” induced by EVA), sensorineural (probably caused by hyperosmotic fluid reflux into the basal end of the cochlea, leading to electrolyte imbalance and subsequent cellular damage), or mixed (due to combined causes) [23].

Vestibular symptoms are often present in these patients, although infrequently investigated, because they are subjectively less relevant than the cochlear ones [28]. Unilateral EVA is not a strictly unilateral process; in fact, hearing loss and vestibular hypofunction have also been observed on the contralateral side, not affected by EVA [29]. In our patient, although conductive hearing loss was present, there was no evidence of ever experiencing any vestibular symptoms, so we decided not to perform an in-depth vestibular function analysis.

Even if imaging is not sufficient to explain still open questions about the above-mentioned conditions (i.e. the pathophysiology of EVA), in our opinion, it should have been performed earlier in this patient.

The conductive component of hearing loss cannot be resolved by stapes surgery, which may still be reasonable for stapes fixation and ossicular anomalies (without concomitant inner ear pathologies). In the literature, satisfactory outcomes are reported if incus continuity and mobility are preserved [30]. Unfortunately, this was not present in our patient, which, on the other hand, could have benefitted from an ossiculoplasty.

In general, patients with EVA are expected to show mixed hearing loss due to inner ear mechanisms, and middle ear surgeries probably can only provide a temporary benefit. Indeed, a great number of patients with EVA present fluctuating hearing loss worsening over time, which requires air-conduction hearing aids or even cochlear implantation.

The previously described functional and anatomical data are crucial for the management of these patients and for the choice among surgical options among different types of hearing rehabilitation.

The patient is currently living a normal life, fully integrated into society. He is scheduled for regular audiological evaluations, with seriate vocal and tonal audiometry, to early detect a possible worsening in his hearing function.

## 4. Conclusions

The combination of a malformation of the ossicular chain along with enlarged vestibular aqueduct (EVA) is an extremely rare condition. Our patient was treated for a long time as though affected by simple conductive hearing loss due to recurrent otitis media, and imaging was performed very late in his clinical history. Indeed, the results of the cone beam computed tomography (CBCT) scan would have been crucial for the improvement of overall clinical management, saving time and unnecessary procedures.

This case demonstrates that clinicians should be aware that a combination of rare conditions might occur, and we should also expect the unexpected.

## Figures and Tables

**Figure 1 children-10-00360-f001:**
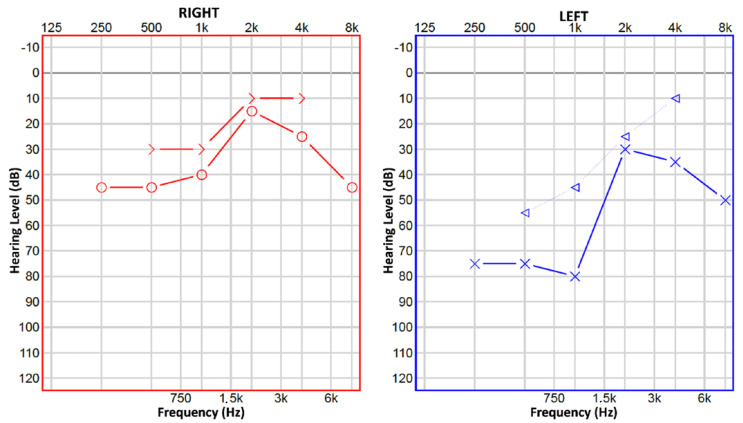
Pure tone audiometry: on the right side, mild−moderate mixed hearing loss, descending on the high and low frequencies. On the left side, mixed hearing loss, severe on the low frequencies, moderate on the high ones. Legend: O: air conduction for right ear, >: bone conduction for right ear, X: air conduction for left ear, △: masked bone conduction for left ear.

**Figure 2 children-10-00360-f002:**
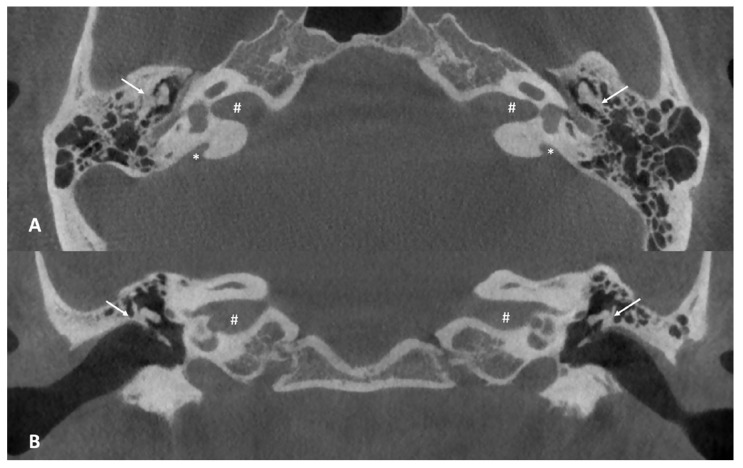
(**A**) Axial cone beam CT scan of the temporal bone. Note the abnormal anatomy of the middle ear, with the ossicular chain almost “fused” with the lateral wall of the tympanic cavity (white arrows) and the enlarged vestibular aqueduct on both sides (white asterisks), 1.74 mm on the right and 2.45 mm on the left. (**B**) Coronal cone beam CT scan of the temporal bone. Note the fusion of the incus to the lateral wall of the tympanic cavity (white arrows). The white hash (#) found in both axial and coronal images, identifies the internal auditory canal.

## Data Availability

Not applicable.

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
