# Peer review of "A Peculiar Case of Ossicular Chain Fixation and Enlarged Vestibular Aqueduct"

_children, 2023, doi:10.3390/children10020360_

Round 1

Reviewer 1 Report

children-2175179: "Otitis media, ossicular chain fixation, vestibular aqueduct..." 

This is a review of a very unfortunate case in which a pediatric patient was misdiagnosed for years and will only now (and made too late) receive the proper surgical care for his hearing. 

I have some very minor questions and comments: Please comment, in the text, as appropriate:

1) Why was the stapedotomy performed on the right when it is implied that the left ear was worse all along. A simple comment around line 97 would be useful. Maybe the authors don't know about the level of asymetry over the patient's life.

2) Why was tympanometry not performed? Or was it?

3) Not being familiar with the Italian health system, I am surprised that a stapedotomy would have been performed without a better work-up. Are these not performed by otologist/neurotologist, who would (or should) have noticed the incus fusion upon entering the middle ear space. Maybe there's not answer here. 

Medium:

L 38: "juvenile otosclerosis, so stapedotomy was performed on his right side": but Fig.1 shows the left ear is a lot worse. Why was the right ear operated on? (and not the left) (sort of the same question as point 1 above)

L42: "the patient was referred to our center to obtain a second opinion": are the authors at a tertiary center? It would be worth spending a sentence on the level of the clinic the authors work at, for readers not familiar with the Italian system (on line 42)

Minor:

title: "elargement" is not a word

L 25:  "worse the left side" 

Author Response

Dear Reviewer, we thank you for your suggestions. Herby our answers:

1) Why was the stapedotomy performed on the right when it is implied that the left ear was worse all along. A simple comment around line 97 would be useful. Maybe the authors don't know about the level of asymmetry over the patient's life.

We do not have access to the pure tone audiometry of the patient before surgery, thus we are not able to say which ear was worse when the doctors decided to perform surgery.

2) Why was tympanometry not performed? Or was it?

The patient was treated in another center before coming to our attention. He did non bring any documentation regarding a previous tympanometry.

3) Not being familiar with the Italian health system, I am surprised that a stapedotomy would have been performed without a better work-up. Are these not performed by otologist/neurotologist, who would (or should) have noticed the incus fusion upon entering the middle ear space. Maybe there's not answer here

Unfortunately we do not have the answer to this question. Usually before undergoing surgery a better work up is needed. We do not have access to the surgery transcription of the stapedoplasty, so we do not know if the surgeon noticed something was amiss in the ossicular chain.

L 38: "juvenile otosclerosis, so stapedotomy was performed on his right side": but Fig.1 shows the left ear is a lot worse. Why was the right ear operated on? (and not the left) (sort of the same question as point 1 above)

Figure 1 shows the audiometric examination performed in our center, in the previous documentation the hearing loss is referred to as bilateral, we do not know why they decided to implant the right side.

L42: "the patient was referred to our center to obtain a second opinion": are the authors at a tertiary center? It would be worth spending a sentence on the level of the clinic the authors work at, for readers not familiar with the Italian system (on line 42)

We added this data in the text.

title: "elargement" is not a word

The title has been changed accordingly

L 25:  "worse the left side"

We revised  it

Reviewer 2 Report

The title doesn't doesn't make sense and/or has typos in it. The introduction is ridiculously short etc.

Author Response

Dear Reviewer, we thank you for your suggestions. Herby our answers:

The title doesn't doesn't make sense and/or has typos in it. The introduction is ridiculously short etc.

We changed the title and implemented the article

Reviewer 3 Report

This manuscript describes the rare case of enlarged vestibular aqueduct and ossicular chain fixation, which will be valuable for the otological clinicians. The following additional data may be of interest to readers:

The course of pre tone audiometry before and after the first surgery if available.

The current treatment plan for this case.

Author Response

Dear Reviewer, we thank you for your suggestions. Herby our answers:

This manuscript describes the rare case of enlarged vestibular aqueduct and ossicular chain fixation, which will be valuable for the otological clinicians. The following additional data may be of interest to readers:

The course of pre tone audiometry before and after the first surgery if available:

Unfortunately the surgery was not performed in our center and we do not have access to those data

The current treatment plan for this case

We added it in the discussion
